# Environmental Tobacco Smoke Exposure in Relation to Family Characteristics, Stressors and Chemical Co-Exposures in California Girls

**DOI:** 10.3390/ijerph16214208

**Published:** 2019-10-30

**Authors:** Gayle C. Windham, Jasmine W. Soriano, Dina Dobraca, Connie S. Sosnoff, Robert A. Hiatt, Lawrence H. Kushi

**Affiliations:** 1Environmental Health Investigations Branch, CA Department of Public Health, 850 Marina Bay Parkway, Richmond, CA 94804, USA; dina.dobraca@cdph.ca.gov; 2Sequoia Foundation, 2166 Avenida de la Playa, Suite D, La Jolla, San Diego, CA 92037, USA; JSoriano@marincounty.org; 3Centers for Disease Control and Prevention, 4770 Buford Hwy NE, Atlanta, GA 30341, USA; css3@cdc.gov; 4Department of Epidemiology and Biostatistics, University of California, 550 16th Street, 2nd floor, Box #0560, San Francisco, CA 94143, USA; robert.hiatt@ucsf.edu; 5Kaiser Permanente Division of Research, 2000 Broadway, Oakland, CA 94612, USA; Larry.Kushi@nsmtp.kp.org

**Keywords:** environmental tobacco smoke exposure, cotinine, second-hand smoke exposure, chemical exposure, environmental justice, children’s health

## Abstract

Childhood environmental tobacco smoke (ETS) exposure is a risk factor for adverse health outcomes and may disproportionately burden lower socioeconomic status groups, exacerbating health disparities. We explored associations of demographic factors, stressful life events, and chemical co-exposures, with cotinine levels, among girls in the CYGNET Study. Data were collected from families of girls aged 6–8 years old in Northern California, through clinic exams, questionnaires and biospecimens (n = 421). Linear regression and factor analysis were conducted to explore predictors of urinary cotinine and co-exposure body burdens, respectively. In unadjusted models, geometric mean cotinine concentrations were higher among Black (0.59 ug/g creatinine) than non-Hispanic white (0.27), Asian (0.32), or Hispanic (0.34) participants. Following adjustment, living in a rented home, lower primary caregiver education, and lack of two biologic parents in the home were associated with higher cotinine concentrations. Girls who experienced parental separation or unemployment in the family had higher unadjusted cotinine concentrations. Higher cotinine was also associated with higher polybrominated diphenyl ether and metals concentrations. Our findings have environmental justice implications as Black and socio-economically disadvantaged young girls experienced higher ETS exposure, also associated with higher exposure to other chemicals. Efforts to reduce ETS and co-exposures should account for other disparity-related factors.

## 1. Introduction

Environmental tobacco smoke (ETS) exposure in children contributes to numerous adverse health outcomes including respiratory infections, ear infections, severe asthma, and impeded lung development [1,2]. Though the prevalence of ETS, also known as passive smoking or second-hand smoke, exposure among children is estimated to have declined in the U.S. by 37% from 1999–2000 to 2011–2012, approximately 15 million 3–11 year olds were still exposed to ETS in 2011–2012, and disparities by race and socioeconomic status (SES) persist [3].

Several studies report ETS exposure is higher among non-Hispanic Black children than other race/ethnicity groups [1,3,4,5,6,7]. In the National Health and Nutrition Examination Survey (NHANES), among children who did not live with smokers, Black children had the highest levels of serum cotinine, a metabolite of nicotine and established biomarker of ETS, compared to all other race/ethnic groups [6]. In contrast, studies consistently show Hispanic children have lower levels of cotinine, compared to other ethnicities [1,5,6,8]. Although nicotine metabolism may vary by race [9,10,11,12], thus contributing to these differences in cotinine levels, they may also result from differential exposures to cigarette smoke.

Indicators of low socioeconomic status (SES), such as family income and parental education, are also established predictors of ETS exposure among children [1,3,4,5,13,14,15,16,17]. In 2007, U.S. children from lower SES households had over seven times the risk of ETS exposure than their higher SES counterparts [8]. Additionally, children living in single-parent households, homes with more residents, and rented homes have been shown to experience a higher prevalence of ETS exposure [3,13,14,18,19,20].

Furthermore, these demographic predictors of ETS exposure have been associated with disproportionate exposure to other harmful environmental exposures, contributing to concerns about environmental justice [1,21]. Exposure to lead, a known neurotoxin, is higher among Black children and those living in poverty than among children of other races and higher income households [1]. From NHANES data, Bisphenol A (BPA) levels were significantly higher among Black children than among other races, while levels of dibutyl phthalate were higher in children living below the poverty level [1]. A study analyzing measures of 75 individual biomarkers among children living in an underprivileged, low-income region of Minnesota revealed that levels of several chemicals including phthalates, organochlorine pesticides, metals, polychlorinated biphenyls, and volatile organic compounds were higher than national averages [22]. Previous analyses from the study cohort examined in the present paper showed that levels of various chemicals tended to be higher in Black compared to white participants, including; all six measured polybrominated diphenyl ethers (PBDEs), as well as some metals and phthalates [23,24,25].

Children’s health disparities are related to SES factors, so hazardous environmental exposures associated with the same patterns of socioeconomic disparities may contribute to higher morbidity and mortality [26]. Stress may be a mediating factor between SES predictors, exposures and health outcomes [21,27], creating a pattern where disadvantaged subpopulations of children are likely disproportionately burdened by multiple harmful risk factors. Children are especially vulnerable to violations of environmental justice tenets, or the unequal distribution of protection from harmful environmental influences [28,29], in part due to greater susceptibility to adverse health impacts as they are still developing [28,30]. Researchers have called for more emphasis on investigating social and environmental risk factors together to better understand implications of environmental inequalities [26,28].

The objective of this investigation was to identify demographic and household predictors of urinary cotinine, as a marker of ETS exposure, in a study of young girls, while also examining whether other stressors, such as stressful life events in the family and other environmental exposures, were associated, thereby identifying them as potential mediators of health effects for future studies. To our knowledge, no research has explored the relationships of demographic, socioeconomic, chemical, and psychosocial stress exposures with ETS together in a sample of young children.

## 2. Materials and Methods

### 2.1. Study Population

The Cohort Study of Young Girls’ Nutrition, Environment, and Transitions (CYGNET) is a prospective cohort study of girls aged 6–8 years at baseline that was focused on studying predictors of the age at onset of puberty and other hallmarks of growth and the pubertal transition. The CYGNET Study was one of three puberty cohort studies of the NIEHS/NCI Breast Cancer and the Environment Research Program (BCERP) and methods have been described previously [23,31,32,33,34]. Study participants were recruited from the membership of the Kaiser Permanente Northern California (KPNC) health care system. Briefly, in 2005–2006, families of girls who were born in and were current members of the KPNC Health Plan in the San Francisco Bay Area were recruited into the study. Eligibility criteria included girls who were 6–8 years old at the time of entry, had no history of endocrine-associated medical conditions, and lived in the San Francisco Bay Area to facilitate in-person exams and biospecimen collection. The final cohort included 444 girls. Baseline interviews were conducted with the primary caregiver and participants were followed annually until age 15, on average. Biomarker data in this analysis were from urine samples collected from all participants at baseline and blood samples collected within the first three years of the study. The study was approved by KPNC’s institutional review board (IRB), with the Centers for Disease Control and Prevention (CDC) IRB approving reliance on the KPNC IRB. Parents completed informed consent initially and for any changes in protocol in subsequent years, while girls provided assent when age-appropriate.

### 2.2. Data Collection

At baseline, the parent or guardian of each participant responded to a detailed questionnaire that included demographics and household characteristics, administered by a trained interviewer. Race was asked open-ended and then coded into multiple pre-coded categories; if more than one race group was reported, the respondent was asked to choose the “best representation,” if possible. Hispanic/Latina ethnicity was asked as a separate question. These were combined hierarchically (and mutually exclusively) as Black (regardless of ethnicity), Hispanic (any race other than Black), Asian or Pacific Islander, and non-Hispanic (N-H) white, so that multi-racial was coded into a single (“highest”) category. Household income was asked as a series of multiple choice questions and responses were categorized for analysis as follows: Less than $50,000, $50,000–$100,000 and $100,000 or greater. Questions were also asked about the number of rooms in the girl’s primary home (including kitchen, living room, dining room, bedrooms, dens and family rooms), and the number of people living in the home and their relationship to the participant. From the responses, we created the following binary variables: living with both biological parents, living with biological siblings, and living in a “crowded” home (≥1 person per room). Questions about smoking among household members (and regular visitors), by relationship and amount smoked, were also asked.

Questions about stressful life events were not asked in the baseline questionnaire, but rather in the parent questionnaire in the ninth annual visit, by which point fewer participants were still being followed (n = 321). Questions were adapted for this study from the ten most stressful life events on the Social Readjustment Rating Scale (SRRS), more commonly known as the Holmes and Rahe Stress Scale [35]. Parents and guardians were asked about whether the family had experienced specific stressful life events during the mother’s pregnancy with the participant or in the first six years of the participant’s life, separately. The stressful life events included were: “*Did you or your husband/partner lose a job, or were unemployed (not by choice) for one month or longer?*”, “*Did you or a close family member (husband/partner, parent, child, sibling) have a serious illness or injury?*”, “*Did a close family member (parent, child, sibling) die?*”, “*Did you get separated, divorced, or experience serious difficulties with your husband/partner?*”, and “*Were there serious legal or financial problems for you or your husband/partner?*”. These were examined individually and summed (0–5), then categorized as 0, 1–2, or ≥3 events during pregnancy or childhood.

### 2.3. Biomarker Assays

Urine samples were collected annually, but our analysis is focused on baseline samples (n = 421 with cotinine), when all biomarkers had been measured. Blood samples were provided in years one and/or two (and a few in year three) (n = 350 girls provided at least one). Urine and blood samples were stored in −80 C freezers. The CDC laboratory analyzed cotinine using high performance liquid chromatography/atmospheric pressure ionization tandem mass spectrometry (LC/MS/MS), previously described [33,36,37]. Using quality control samples, cotinine measurements were determined to be in statistical control according to standard criteria [38]. Cotinine data, reported in nanograms per milliliter, were available for 421 girls, of whom 96% had detectable values (limit of detection (LOD) of 0.036 ng/mL). Levels below the LOD were imputed as LOD/√2. Creatinine was measured by a colorimetric enzymatic assay on a commercial automated clinical chemistry analyzer. Cotinine values were corrected for creatinine to account for urine concentration.

Urine and blood/serum samples were tested for several other biomarkers as previously reported. In addition to cotinine, this analysis used data from ten categories of biological analytes: urinary metals [24], urinary polycyclic aromatic hydrocarbons (PAHs) [39], urinary phthalates and phenol metabolites [25,40], blood metals [41], and serum per- and poly-fluoroalkyl substances (PFAS) [42], polybrominated diphenyl ethers (PBDEs), polychlorinated biphenyls (PCBs), and organo-chlorinated pesticides (OCPs) [23,34]. The latter three serum concentrations were lipid-adjusted (nanograms per gram lipid weight), with lipids determined using commercially available kits (Roche Diagnostics Indianapolis, IN, 46256, USA.) for total triglycerides and total cholesterol. Urinary biomarkers were creatinine-corrected (metals, phenols, phthalates, PAHs). Individual analytes/congeners within these groups are listed in Appendix A. Congeners that were below the LOD in more than 40% of the samples were not considered for further analysis; these included urinary beryllium, cadmium, platinum, 4-OH-phenanthrene, and blood cadmium. For the remainder, levels below the LOD were imputed as LOD/√2.

### 2.4. Statistical Analysis

All participants with valid cotinine values were included in the analysis dataset (N = 421). The distribution of characteristics was examined by creatinine-adjusted cotinine quartiles, with a chi-square test to assess significant differences. These characteristics included: child’s race, household income, caregiver education, home ownership, two parents in the home, siblings in home, crowding, and maternal age at child’s birth. Unadjusted linear regression was conducted to assess whether demographic and household variables predicted the continuous natural log of creatinine-adjusted urinary cotinine. Geometric means (GMs) and their 95% confidence intervals (CI) were also calculated.

For adjusted models, we included the demographic and household variables and examined the variance inflation factor to assess collinearity. Maternal age was collinear with other independent variables in the model so we excluded it from the fully adjusted model. Stepwise selection (significance level of entry = 0.50, significance level of staying = 0.05) was used to produce a “reduced” model. Predicted adjusted GMs were calculated using generalized linear models assuming the distribution of characteristics in the dataset. To assess whether race/ethnicity modifies the association between household characteristics and cotinine, we conducted stratified analyses including the independent variables from the reduced model due to smaller sample sizes. We examined whether stressful life events predicted cotinine concentrations using unadjusted linear regression and models adjusting for child’s primary caregiver educational attainment and home ownership.

We conducted a sensitivity analysis to determine whether predictors of cotinine were driven by demographic differences related to smoking (parents). The primary analyses were re-run restricted to girls who did not live in a household with smokers, regardless of whether they smoked in the home or not, as numbers were small (n = 80 smoking homes), this may be misreported, or third-hand smoke exposure could occur.

We initially examined the associations of each chemical and chemical group with cotinine, using correlations and modelling GM cotinine levels by chemical quartiles. To consider all inter-relationships together we conducted a factor analysis with all the chemicals/congeners that met detection criteria, using a varimax rotation. Chemicals with loading factors of greater than 0.35 or less than −0.35 were listed as included in the factor. The seven-factor model was selected because of the number and strength of loading factors. Unadjusted linear regression models were run with the resulting factors predicting cotinine concentrations. Potential confounders were not included because the goal of the analysis was to assess whether participants more highly exposed to ETS were also disproportionately exposed to other chemical groups.

## 3. Results

Participants’ families were generally of high SES as indicated by 42% having a family income of at least $100,000, 72% owning their home, and 51% of primary caregivers having a Bachelor degree (Table 1). At baseline, the majority of participants had two parents (81%) and siblings (77%) in the household, and did not live in a crowded home (91%). The cohort was diverse with a distribution of 22% Black, 24% Hispanic, 12% Asian/PI, and 42% N-H white. Overall, 72% of participants with cotinine measured remained in the cohort for the ninth annual exam and responded to questions about stressful life events; of those, 32% of families experienced at least one event during pregnancy, and 67% experienced at least one event from infancy to 6 years old (Table 1).

### 3.1. Predictors of Cotinine Levels

The distribution of cotinine varied significantly by race. Black participants were more likely to have creatinine-adjusted cotinine concentrations in the highest quartile (45%), while white participants were the least likely (17%) (Table 1). Participants were also more likely to have cotinine concentrations in the highest quartile if: their family was in the lowest income level (46%) compared to the highest (16%); their family did not own their home (47% vs. 17% owning their home); they lived in a crowded home (44% vs. 23%); they did not have two biologic parents in their primary home (56% vs. 18%); or they had no siblings in the household (35% vs. 22% with at least one sibling). Cotinine levels varied with caregiver educational attainment, however not in a monotonic direction. Participants whose primary caregiver had some college education were more likely to be in the highest cotinine quartile (42%) than participants whose caregiver had lower (31%), or higher (12%) education. Occurrence of stressful life events during the pregnancy was not associated with cotinine quartile, and the pattern of higher exposure with three or more events versus none during early childhood (35% vs. 17%) was not statistically significant. However, comparing three or more events during early childhood to fewer was borderline statistically significant (*p* = 0.05).

Table 2 presents unadjusted and adjusted geometric means of creatinine-adjusted cotinine by demographic and household characteristics. Consistent with results by quartile, cotinine levels were higher with lower family income, primary parent education of some college versus high school or Bachelor degree, not owning a home, not having two parents in the household, not having siblings in the household, and having a crowded home. Cotinine levels were highest in Black participants (GM: 0.59 ug/g creatinine, 95% CI: 0.43, 0.81) compared with Asian (GM: 0.32, 95% CI: 0.23, 0.44), Hispanic (GM: 0.34, 95% CI: 0.28–0.42), and N-H white (GM: 0.27, 95% CI: 0.22, 0.32) participants. Differences in geometric means by race/ethnicity, family income, crowding and presence of siblings were attenuated after adjustment for all other factors (Table 2), leaving the caregiver education, home ownership and two parents in the household variables as significant.

Stratifying by race/ethnicity with reduced adjustment models (Table 3), home ownership was the most predictive of cotinine GM (lower levels than among renters), across all groups except Asians, and was the only significant predictor of cotinine among Black and Hispanic participants. Among N-H white participants, lower primary caregiver education (not completing college vs. bachelor’s degree or higher) and not having two parents in the home also predicted higher cotinine concentrations (*p* < 0.01 for all).

There were differences between the subgroup of girls who were still in the study to have stressful life events questions answered and those who were not available; they had higher family income, higher primary caregiver education and fewer smokers in the home (*p* = 0.05). Furthermore, they had somewhat lower cotinine levels than those who did not have stressful life events responses (GMs: 0.32 ug/g creatinine vs. 0.40, respectively, *p* = 0.12). Linear regression models showed few associations between number of stressful life events either during pregnancy or in early childhood (0 to 6 years old) and the natural log of urinary creatinine-adjusted cotinine (Table 4). In unadjusted models, participants who experienced separation or divorce of parents in childhood had higher cotinine levels compared with those who did not (0.42 vs. 0.30 ug/g creatinine, *p* = 0.03), as did those that experienced unemployment in the family (0.42 vs. 0.30, *p* = 0.04). These associations did not remain significant after adjustment for socio-demographic factors.

As expected, some characteristics varied between girls who lived in homes with smokers versus those who did not; they had higher cotinine levels (GM 1.4 vs. 0.25 ug/g creatine, *p* < 0.001), were more likely to be Black, with lower family income, lower caregiver education, less likely to have two parents in the home, or more likely to live in a rented home (data not shown). In addition, they were more likely to experience 3 or more stressful life events in utero or during early childhood (10% vs. 2% and 33% vs. 19%, respectively, *p* <0.05). Restricting analyses to girls living in “non-smoker” homes yielded very similar patterns of results for the relationships of GM cotinine levels with demographic factors as overall (Appendix A), with living in a rented home, not having two parents in the household, and caregiver education remaining significantly associated with higher GMs after adjustment for all other factors. The pattern of relationship between stressful life events and cotinine was also relatively similar to overall results, with however, unemployment in the family during the girl’s childhood remaining a significant (*p* < 0.02) predictor of cotinine level after adjustment (Appendix A).

### 3.2. Chemical Co-Exposure Analysis

Many of the chemical biomarkers were associated with cotinine, either positively or negatively, with the PFAS and phthalate metabolite groups least likely to include any associated compounds. Examining seven factors indicating interrelationships between co-exposures and cotinine (Table 5), we observed that factors loaded heavily on congeners within the same chemical category or categories. For example, Factor 1 comprised 12 PCBs and three OCPs and Factor 2 loaded heavily with seven PBDEs. Using the factors to predict cotinine in linear regression models, PBDE concentrations (Factor 2) were positively associated with cotinine (*p* < 0.01), as were selected urinary metal (i.e., tungsten, uranium, cobalt, manganese, barium and lead) concentrations (Factor 5) (*p* = 0.02). Additionally, Factor 7 predicted cotinine (*p* = 0.02), with two phenols, blood and urinary lead positively associated, and blood mercury and arsenic negatively associated. There was also some suggestion of a negative association with cotinine with Factor 1, though not statistically significantly.

## 4. Discussion

The present study identified demographic (i.e., race/ethnicity) and socioeconomic (i.e., caregiver education, home ownership, lack of two parents in the home, family income) predictors of ETS exposure, assessed by urinary cotinine, in young girls in the San Francisco Bay Area. Consistent with previous research [1,3,4,5,13,14,15,16,17], our results indicate social factors that are generally associated with lower SES predict greater ETS exposure. The importance of living in a rented home as a risk factor for ETS exposure was further emphasized in race-stratified analyses, where it remained an independent predictor even when adjusted for education and two parents in the home, among Black, Hispanic and N-H white participants. Some of these factors may reflect demographics of smokers; however, the associations of ETS with living in a rental home and lack of two biologic parents in the home were still apparent among the girls in homes without regular smokers.

Rental homes may be more likely to include multi-family dwellings in which individuals have less control over current ETS exposures or even third-hand smoke exposure from prior residents. Analyses from the New York site of BCERP revealed that children in public housing had higher cotinine levels than those in private housing, even when accounting for the presence of a smoker in the home [43]. Additionally, our results suggest other characteristics including low primary caregiver education and not living with two parents are predictive among N-H whites. Results indicate that interventions to reduce ETS could be beneficially targeted toward more vulnerable communities or families of lower SES or that are renting homes. By the beginning of the study, California had banned smoking in the workplace (Cal. Lab. Code §6404.5), in playgrounds (Cal. Bus & Prof. Code §22961), in daycare facilities (Cal. Health & Safety Code §1596.795), and within 20 feet of an entrance of a public building (Cal. Gov. Code §7597(a)). The association with renting warrants further investigation and consideration from policymakers, or enforcement of current policies.

Examining adverse “co-exposures” (or stressors), participants who experienced unemployment in the family or parental separation, consistent with the finding of lack of two biologic parents in the household as a predictor, had higher mean cotinine levels. This suggests children may experience certain types of stressful life events and hazardous ETS exposure simultaneously, potentially exacerbating health effects, which we did not examine in this analysis. Perhaps a family not owning their home may also represent a stressor in terms of moving more frequently and less home security/safety. Furthermore, these results could underestimate effects if associations were attenuated by selection bias. The subgroup who responded to stressful life events questions was generally higher SES and less likely to have a smoker in the home than those missing these questions, but creatinine-adjusted cotinine was not significantly different. These questions were derived from standardized questionnaires used in other health research, but may be less recalled further from the events, leading to possible misclassification, however this would seem unlikely to be biased by cotinine level. These do not represent all possible psychosocial stressors, such as would be included in an Adverse Childhood Experiences (ACE) assessment [44]. Asking about ACEs was beyond the scope of the study and may not have been reported accurately by caregivers due to their sensitive nature.

Higher co-exposure to polybrominated diphenyl ethers (PBDEs), several metals and some phenol metabolites was associated with higher ETS exposure as well. Tobacco smoke itself represents a mixture of hundreds of chemicals, so these may comprise some of the “co-exposure”, particularly for metals [45]. The co-exposures we examined generally represent multiple sources, such as food and its preparation, personal care products, and environmental contamination. Together with the finding that lower SES participants have higher cotinine levels, these results indicate environmental exposures with potential adverse health effects cluster in vulnerable populations. From an environmental justice perspective, this confirms there are inequities in which disadvantaged populations may experience multiple exposures that may negatively impact health [21]. Efforts to reduce these social inequities could have multiple salutary effects.

There are limitations to the present research. The sample size restricted the number of comparisons that could be made within analyses stratified by race/ethnicity and in the subset with stress questions. The participants included girls only, residing in California and from families of somewhat high SES, so are not representative of the general U.S. population or other countries. Most studies have not found cotinine differences related to gender/sex [46,47,48]. California has been progressive about lowering smoke exposures, so that children were much less likely to reside with someone smoking in the home than in other states in a 2007 national survey [8]. Within the BCERP, the geometric mean urinary cotinine level was 4–5 times lower in our N. CA site than those in Cincinnati or New York City and about half what we estimated for NHANES U.S. data [33]. In an Italian study [46] conducted in slightly later years, the median cotinine reported in children not living with smokers was 1.8 ug/g creatinine, considerably higher than our overall median of 0.26 ug/g creatinine. Despite the low levels, we were able to identify significant predictors. Additionally, a single urinary cotinine measurement may not be representative of regular ETS since it reflects exposure to tobacco smoke within a relatively short duration. However, previous analyses from the same study showed that urinary cotinine levels were associated with questionnaire assessments of ETS exposure, as well as high repeatability of cotinine concentrations across a few years, as indicated by moderate to substantial agreement in intra-class correlations [33]. While we used cotinine as a measure for ETS exposure, cotinine levels also reflect direct tobacco use. However, as would be expected from children ages 6–8 years, only one participant had baseline urinary cotinine concentration greater than 50 ng/mL, a level recommended by the lab as a conservative upper limit of ETS exposure. This study focused on household and family characteristics predicting cotinine levels. However, because ETS exposure is not limited to the home, characteristics related to activities away from the home, neighborhood, and associated lifestyle factors should be further included in studies of determinants of ETS and other exposures.

There are several strengths to this study. The CYGNET study used rigorous epidemiologic methods throughout the sample selection, data collection, data management and biomarker assay. The detailed questionnaire data about demographics, household factors, and stressful life events allowed us to examine numerous possible predictors, not included together in other studies, to our knowledge. The use of urinary cotinine as a measure of ETS exposure is not subject to reporting misclassification. Further a multitude of other exposures were measured in biospecimens, providing a unique opportunity. Using this biomarker data, we examined associations of cotinine with other chemicals to assess the potential for an accumulation of environmental exposures within potentially EJ-impacted subpopulations. The results could help guide ETS exposure prevention efforts and shed insight into the pattern of multiple exposures among vulnerable populations.

## 5. Conclusions

This analysis shows that cotinine concentrations (as a measure of ETS exposures) among young girls in the San Francisco Bay Area are higher among groups with certain demographic and household characteristics that are more common in vulnerable populations who may also be experiencing stressful life events. Furthermore, children who are exposed to ETS appear more likely to be exposed to other chemicals, particularly brominated flame retardants and heavy metals, emphasizing concerns about effects of chemical mixtures on health. These findings support further examination of the burden of health impacts among children with multiple exposures and daily stressors and ways to ameliorate them.

## Figures and Tables

**Table 1 ijerph-16-04208-t001:** Demographic and Household Structure Characteristics by Quartile of Urinary Cotinine.

Characteristics	% Participants (N = 421)	Q1 ^1^ N (%) ^2^	Q2 N (%)	Q3 N (%)	Q4 N (%)	χ^2^ *p*-Value
Race						<0.01
Asian	12	13 (25)	15 (29)	13 (25)	11 (21)	
Black	22	18 (20)	16 (17)	17 (18)	41 (45)	
Hispanic	24	21 (21)	28 (27)	29 (28)	24 (24)	
White	42	54 (31)	46 (26)	46 (26)	29 (17)	
Family Income						<0.01
<$50 k/yr	21	12 (14)	15 (17)	20 (23)	40 (46)	
≥$50, <100 k	37	34 (22)	41 (27)	44 (29)	35 (23)	
≥$100 k/yr	42	58 (33)	48 (28)	40 (23)	28 (16)	
Primary Caregiver Education						<0.01
College Grad (B’s)	51	70 (33)	57 (27)	60 (28)	25 (12)	
Some College	31	21 (16)	32 (25)	22 (17)	54 (42)	
≤High School (HS)	19	15 (19)	16 (21)	23 (29)	24 (31)	
Home Ownership						<0.01
Is Renter	28	14 (12)	18 (16)	30 (26)	54 (47)	
Owns Home	72	92 (30)	86 (28)	75 (25)	50 (17)	
Two Parents in Household ^3^						<0.01
Yes	81	96 (28)	94 (27)	92 (27)	61 (18)	
No	19	10 (13)	11 (14)	13 (17)	44 (56)	
Siblings in Household ^4^						<0.01
Yes	77	82 (25)	92 (28)	78 (24)	71 (22)	
No	23	24 (24)	13 (13)	27 (28)	34 (35)	
Crowding (# people/# rooms)						0.01
Not Crowded (<1)	91	97 (25)	102 (27)	95 (25)	88 (23)	
Crowded (≥1)	9	9 (23)	3 (8)	10 (26)	17 (44)	
Stressful Life Events during Pregnancy ^5^						0.94
None	68	52 (25)	59 (29)	49 (24)	45 (22)	
1–2 events	29	21 (24)	21 (24)	25 (28)	21 (24)	
3 or more events	3	2 (20)	2 (20)	3 (30)	3 (30)	
Stressful Life Events from 0 to 6 years ^5^						0.19
None	33	29 (29)	28 (28)	27 (27)	17 (17)	
1–2 events	45	33 (24)	40 (29)	35 (26)	29 (21)	
3 or more events	22	13 (20)	14 (22)	15 (23)	23 (35)	

^1^ Quartile 1 (Q1): 0.017–0.151 µg/g creatinine; Q2: 0.152–0.259 µg/g creatinine, Q3: 0.260–0.592 µg/g creatinine, Q3: 0.593–89.212 µg/g creatinine. ^2^ Row percent. ^3^ Parent was considered a biological parent. ^4^ Sibling was considered a biological sibling. ^5^ Stressful life events assessed at 9th visit among n = 303 with cotinine.

**Table 2 ijerph-16-04208-t002:** Urinary Cotinine (µg/g creatinine) Geometric Means (GM) by Demographic and Household Characteristics.

Characteristics		Unadjusted (N = 421)			Adjusted ^1^ (N = 412)		
	%	GM	95% CI	*p*	GM ^2^	95% CI	*p*
Race							
Asian	12	0.32	(0.23–0.44)	0.35	0.33	(0.24–0.45)	0.74
Black	22	0.59	(0.43–0.81)	<0.01	0.42	(0.32–0.54)	0.28
Hispanic	24	0.34	(0.28–0.42)	0.11	0.27	(0.21–0.35)	0.13
N-H White	42	0.27	(0.22–0.32)	ref	0.35	(0.29–0.42)	ref
Family Income							
>=$100 k/yr	42	0.24	(0.21–0.28)	ref	0.32	(0.27–0.39)	ref
>=$50, <100 k/yr	37	0.35	(0.28–0.42)	<0.01	0.34	(0.28–0.40)	0.76
<$50 k/yr	21	0.65	(0.48–0.88)	<0.01	0.37	(0.28–0.50)	0.47
Primary Caregiver Education							
College Grad (B’s)	51	0.24	(0.21–0.28)	ref	0.27	(0.23–0.33)	ref
Some College	31	0.51	(0.40–0.65)	<0.01	0.42	(0.34–0.52)	<0.01
≤High School (HS)	19	0.44	(0.33–0.61)	<0.01	0.41	(0.31–0.55)	0.03
Home Ownership							
Owns Home	72	0.26	(0.23–0.30)	ref	0.29	(0.25–0.33)	ref
Is Renter	28	0.69	(0.53–0.90)	<0.01	0.52	(0.42–0.66)	<0.01
Two Parents in Household ^3^							
Yes	81	0.29	(0.26–0.32)	ref	0.31	(0.28–0.35)	ref
No	19	0.75	(0.54–1.03)	<0.01	0.48	(0.36–0.65)	0.01
Siblings in Household ^4^							
Yes	77	0.32	(0.28–0.36)	ref	0.32	(0.29–0.37)	ref
No	23	0.44	(0.33–0.58)	0.03	0.39	(0.31–0.50)	0.16
Crowding (# ppl/# rooms)							
Not Crowded (<1)	91	0.32	(0.29–0.37)	ref	0.33	(0.30–0.37)	ref
Crowded ≥1)	9	0.61	(0.36–1.03)	<0.01	0.42	(0.29–0.62)	0.23

^1^ Adjusted for all listed independent variables. ^2^ Geometric means were calculated using least squares means. ^3^ Parent was considered a biological parent. ^4^ Sibling was considered a biological sibling.

**Table 3 ijerph-16-04208-t003:** Demographic and Household Characteristics Predicting Urinary Cotinine (µg/g creatinine) Geometric Means ^1^, by Race/ethnicity.

	Asian (N = 51)			Black (N = 91)			Hispanic (N = 102)			N-H White (N = 173)		
	GM	95% CI	*p*	GM	95% CI	*p*	GM	95% CI	*p*	GM	95% CI	*p*
Education												
≥B’s	0.24	(0.17–0.34)	ref	0.52	(0.31–0.90)	ref	0.34	(0.22–0.53)	ref	0.22	(0.18–0.26)	ref
Some College	0.63	(0.34–1.16)	<0.01	0.62	(0.42–0.93)	0.61	0.39	(0.27–0.55)	0.68	0.39	(0.28–0.56)	<0.01
≤HS	0.27	(0.12–0.62)	0.83	0.67	(0.26–1.78)	0.67	0.31	(0.23–0.42)	0.71	0.62	(0.37–1.05)	<0.01
Home Ownership												
Owns	0.29	(0.21–0.40)	ref	0.39	(0.25–0.61)	ref	0.26	(0.20–0.34)	ref	0.24	(0.20–0.28)	ref
Rents	0.37	(0.18–0.73)	0.54	0.97	(0.59–1.58)	0.01	0.50	(0.36–0.69)	<0.01	0.51	(0.34–0.76)	<0.01
Two Parents in Home ^2^												
Yes	0.33	(0.24–0.44)	ref	0.46	(0.31–0.71)	ref	0.31	(0.25–0.39)	ref	0.25	(0.21–0.29)	ref
No	0.15	(0.06–0.38)	0.12	0.80	(0.50–1.29)	0.11	0.47	(0.30–0.75)	0.11	0.65	(0.35–1.21)	<0.01

^1^ Linear regression models adjusted for education (B’s is at least a Bachelor’s degree level, HS = high school or less), home ownership, and two parents in the home, predicting natural log of urinary creatinine-adjusted cotinine. ^2^ Parent was defined as a biological parent.

**Table 4 ijerph-16-04208-t004:** Stressful Life Events in Relation to Urinary Cotinine Levels (µg/g creatinine).

Stressful Life Events	N = 321	Unadjusted (N = 303)			Adjusted ^1^ (N = 300)		
	%	GM ^2^	95% CI	*p*	GM ^2^	95% CI	*p*
Pregnancy							
Total # of Stressful Life Events							
None	68	0.32	(0.27–0.38)	ref	0.33	(0.28–0.38)	ref
1–2 events	29	0.33	(0.26–0.41)	0.87	0.31	(0.24–0.39)	0.68
3 or more events	3	0.38	(0.17–0.85)	0.66	0.35	(0.18–0.69)	0.83
Unemployment in the family							
No	92	0.32	(0.28–0.37)	ref	0.32	(0.28–0.37)	ref
Yes	8	0.38	(0.23–0.64)	0.48	0.32	(0.20–0.49)	0.96
Illness or injury in the family							
No	90	0.33	(0.28–0.38)	ref	0.32	(0.28–0.37)	ref
Yes	10	0.31	(0.20–0.46)	0.78	0.31	(0.21–0.46)	0.84
Death of close family member							
No	91	0.32	(0.28–0.37)	ref	0.32	(0.28–0.36)	ref
Yes	9	0.32	(0.21–0.50)	0.99	0.35	(0.23–0.53)	0.70
Separation or divorce							
No	89	0.32	(0.27–0.36)	ref	0.32	(0.28–0.36)	ref
Yes	11	0.40	(0.27–0.57)	0.29	0.34	(0.24–0.50)	0.73
Legal or financial problems							
No	92	0.32	(0.28–0.37)	ref	0.33	(0.29–0.37)	ref
Yes	8	0.32	(0.19–0.52)	0.91	0.29	(0.18–0.44)	0.58
**Childhood (0–6 years)**							
Total # of Stressful Life Events							
None	34	0.30	(0.24–0.38)	ref	0.33	(0.26–0.40)	ref
1–2 events	44	0.31	(0.26–0.38)	0.78	0.31	(0.26–0.37)	0.74
3 or more events	21	0.40	(0.30–0.53)	0.12	0.34	(0.26–0.45)	0.77
Unemployment in the family							
No	77	0.30	(0.26–0.35)	ref	0.31	(0.27–0.35)	ref
Yes	23	0.42	(0.32–0.55)	0.04	0.37	(0.29–0.49)	0.19
Illness or injury in the family							
No	72	0.34	(0.29–0.40)	ref	0.34	(0.30–0.40)	ref
Yes	28	0.28	(0.22–0.37)	0.22	0.28	(0.22–0.35)	0.12
Death of close family member							
No	57	0.32	(0.27–0.38)	ref	0.32	(0.27–0.38)	ref
Yes	43	0.33	(0.27–0.40)	0.82	0.32	(0.27–0.39)	0.94
Separation or divorce							
No	77	0.30	(0.26–0.35)	ref	0.31	(0.27–0.36)	ref
Yes	23	0.42	(0.32–0.56)	0.03	0.35	(0.27–0.46)	0.49
Legal or financial problems							
No	81	0.32	(0.27–0.37)	ref	0.33	(0.29–0.38)	ref
Yes	19	0.35	(0.26–0.48)	0.52	0.29	(0.22–0.39)	0.49

^1^ Adjusted for primary caregiver education and home ownership. ^2^ Geometric means (GM) and 95% confidence intervals were calculated using least squares mean.

**Table 5 ijerph-16-04208-t005:** Biomarker Co-Exposures and Associations with Urinary Cotinine Using Factor Analysis.

Factor	Chemical Class ^1^	Analyte/Congener ^2^	Factor Loading	Beta ^3^	*p* ^3^
1				−0.11	0.15
	PCB	PCB146	0.94		
	PCB	PCB153	0.90		
	PCB	PCB156	0.90		
	PCB	PCB138_158	0.89		
	PCB	PCB187	0.89		
	PCB	PCB99	0.87		
	PCB	PCB170	0.86		
	PCB	PCB74	0.85		
	PCB	PCB180	0.84		
	PCB	PCB118	0.77		
	OCP	OXYCHLOR	0.76		
	PCB	PCB196_203	0.74		
	OCP	T_NONA	0.73		
	OCP	HCB	0.64		
	PCB	PCB105	0.63		
2				0.25	<0.01
	PBDE	PBDE47	0.93		
	PBDE	PBDE100	0.93		
	PBDE	PBDE154	0.91		
	PBDE	PBDE85	0.90		
	PBDE	PBDE99	0.87		
	PBDE	PBDE28	0.78		
	PBDE	PBDE153	0.65		
3				0.12	0.12
	PAH	phen_1	0.92		
	PAH	phen_3	0.91		
	PAH	phen_2	0.87		
	PAH	fluor_9	0.83		
	PAH	fluor_3	0.80		
	PAH	fluor_2	0.76		
	PAH	pyr_1	0.74		
4				0.07	0.35
	Phthalate	mEOHP	0.97		
	Phthalate	mEHHP	0.96		
	Phthalate	mECPP	0.95		
	Phthalate	mEHP	0.93		
5				0.17	0.02
	urinary metal	tungsten	0.75		
	urinary metal	uranium	0.74		
	urinary metal	cobalt	0.62		
	urinary metal	manganese	0.50		
	urinary metal	barium	0.45		
	urinary metal	lead	0.44		
6				−0.02	0.75
	PFAS	PFOS	0.75		
	PFAS	PFOA	0.68		
	PFAS	PFHxS	0.62		
	PFAS	PFDeA	0.46		
	PFAS	PFNA	0.41		
7				0.18	0.02
	Phenol	M_PB	0.64		
	Phenol	P_PB	0.62		
	blood metal	lead	0.41		
	urinary metal	lead	0.38		
	blood metal	mercury	−0.36		
	blood metal	arsenic	−0.43		

^1^ PCB, polychlorinated biphenyls; OCP, organochlorine pesticides; PBDE, polybrominated diphenyl ethers; PAH, polycyclic aromatic hydrocarbons; PFAS, per- and poly-fluoroalkyl substances. See Appendix A for definition of individual compounds. ^2^ Listed analytes have absolute factor loadings of [0.35] or greater. ^3^ Estimates from unadjusted linear regression analysis predicting natural log of creatinine-adjusted urinary cotinine levels.

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
