# Peer review of "Environmental Tobacco Smoke Exposure in Relation to Family Characteristics, Stressors and Chemical Co-Exposures in California Girls"

_ijerph, 2019, doi:10.3390/ijerph16214208_

Round 1

Reviewer 1 Report

General comment:

This study reports interesting data on the relation between some demographic (i.e., race/ethnicity) and socioeconomic (i.e., family income, caregiver education, home ownership, lack of two parents in the home) predictors of SHS exposure, assessed by urinary cotinine, based on a sample of young girls in the San Francisco Bay Area. The methods are well described, but some detail could be added. The statistical analysis is adequate. English is well written.

Specific comments:

Introduction

The introduction is clear, accurate and well referenced. Although not mentioned, taking into account that the sample is constituted only by girls, are there any study focusing specifically on the associations between SHS exposure and the studied variables, that take gender into account? This could be important data to reflect on at least in the Discussion section.

Methods

It is stated in the Study Population that it study consists in a “cohort study of girls aged 6-8 years at baseline”. After it is mentioned that “eligibility criteria included girls who were 6-7 years old at the time of recruitment”. Although I can understand that it is possible to recruit the participants with 7 years and that evaluation can prolong until their 8 years old, this may not be automatically obvious. How was race and ethnicity of the participants collected? Was it by self-report and self-identification with a determined ethnicity? The categories were all previously defined or was there the chance to add any new category, if the participants did not identified with the previously defined ones? It would be relevant to add more information about this in the Methods section, considering this is a sensitive variable. How were the categories defined?

Results

Did the researchers had also data from parents, that could allow to do a correlation analysis between cotinine of parents and children? The authors mention that questions about stressful life events were not asked in the baseline questionnaire, but rather in the parent questionnaire in the ninth annual visit. Could there be any memory bias involved? If the authors had urine samples collected annually, why the option to analyse only the baseline samples?

Reviewer 2 Report

General comments

The manuscript describes the results of a study conducted in California, USA, aimed to identify socio-demographic and household predictors of urinary cotinine levels in young Californian girls (6-8 y.o.). Authors consider u-cotinine as a marker of SHS exposure. Besides, Authors took into consideration other stressors (family events and other environmental exposures).

The manuscript is well written in English, clear and concise; the topic is of interest for the scientific community and for IJERPH. The data obtained are of good relevance for the scientific community and the methodology is interesting.

However, the Authors are not updated on “passive smoking exposure”.

Indeed, passive smoking, actually defined ETS, is generated by two phenomena, the secondhand smoke (SHS) and the thirdhand smoke (THS):

-           SHS is “the combination of smoke emitted from the burning end of a cigarette or other tobacco products and smoke exhaled by the smoker”:

-           THS, which is the residue from tobacco smoke that persists on the clothing and hair of smokers, on environmental surfaces, and in dust long after a cigarette has been extinguished. It is released to the indoor air and inhaled even after days from active smoking.

The passive exposure to SHS occurs when a smoker is smoking, while the passive exposure to THS occurs in the absence of concurrent smoking. Thus, u-cotinine in young girls is a marker of ETS exposure and not only of SHS exposure

In light of this evidence, Authors should modify the introduction section and the rest of the manuscript accordingly, as it is no longer appropriate to use the term “secondhand smoke” as a synonym for passive smoking or for environmental tobacco smoke, because it represents a pars pro toto. In other words, using the term “secondhand smoke” mistakes one part of the problem for the whole.

Specific comments

Title

Environmental Tobacco Smoke instead of SHS

Abstract

To be modified according to my general comments on ETS and SHS/THS

Key words, Highlights, Capsule

Key words: SHS should be modified in ETS

Highlights: not present

Capsule: not present

Introduction

To be modified according to my general comments on ETS and SHS/THS.

Materials and methods

Ok

Results

To be modified according to my general comments on ETS and SHS/THS.

Discussion

To be modified according to my general comments on ETS and SHS/THS.

Besides, the discussion section lacks comparison with data obtained from other researchers on ETS-exposure in children (Protano et al. Urinary levels of trace elements among primary school-aged children from Italy: The contribution of smoking habits of family members. Science of the Total Environment 557-558: 378-385).

Conclusions

To be modified according to my general comments on ETS and SHS/THS.

References

Authors should update the reference list, as suggested

Tables

To be modified according to SHS and ETS meanings

Figures

Not present

Round 2

Reviewer 2 Report

The manuscript can be accepted in the present form (R1 revision)